# Clinical Trials and Therapeutic Approaches for Healthcare Challenges in Pakistan

**DOI:** 10.3390/jpm13111559

**Published:** 2023-10-30

**Authors:** Aamir Ahmed, Norman R. Williams

**Affiliations:** 1ONCOLODYNE Ltd., 71–75 Shelton Street, Covent Garden, London WC2H 9JQ, UK; 2Cell and Developmental Biology, University College London, Gower Street, London WC1E 6JJ, UK; aamir.ahmed@ucl.ac.uk; 3UCL Division of Surgery & Interventional Science, 3rd Floor, Charles Bell House, 43–45 Foley Street, London W1W 7TY, UK

**Keywords:** Pakistan, healthcare costs, health facilities

## Abstract

Pakistan faces tremendous challenges in providing healthcare due to a lack of consistent policymaking, increasing expenditure and exponential growth in population since its inception in 1947. These challenges are not just driven by politics, policy and allocation of resources but also by healthcare, environment and characteristics of the population biology. Clinical trials provide the best way to find population-specific, cost-effective treatments that do not merely mimic those used in wealthier nations. This article analyzes all clinical studies conducted with at least one site in Pakistan listed on ClinicalTrials.gov, combined with a short overview that considers new therapeutic approaches that can be investigated in future clinical trials. Therapies using repurposed medicines are of particular interest as they use affordable drugs that are already widely available.

## 1. Introduction

Since its creation 76 years ago with a nominal infrastructure base, Pakistan has made remarkable progress including improvements in health services. However, it also faces tremendous challenges due to the rapid increase in population since the 1970s, which is currently the fifth largest in the world combined with a modest per capita gross domestic product (GDP, 168th in the world) [1]. This economic deficiency feeds back into health service accessibility, particularly for a large segment of the population that lives at or below the poverty line [2].

These economics reflect in the poor health outcomes compared to developed nations, but they also compare poorly when compared to other low- and middle-income countries (LMICs) [3]. These challenges for Pakistan appear chronic and rather than focusing on societal changes that will require decades, if not centuries, we think innovative solutions are required. The economic challenges to healthcare are further complicated by very high dependence on international, particularly developed world research and testing, necessitating import, associated costs and making therapies inaccessible to most of Pakistan’s populace.

Therefore, there is a specific need for LMICs such as Pakistan to have cost-effective therapies. Coupled with unique cultural, geographic and demographic factors, a critical unmet scientific need is for healthcare treatments that are tested locally for biological and efficiency optimization.

We are interested in developing population-specific, cost-effective therapies particularly targeted towards developing nations. Clinical trials, particularly when randomized, are the best way to discover therapies that are safe and efficacious. These are needed to cover the significant gaps in clinical trial research in Pakistan compared to other neighboring LMICs. For example, in an analysis of breast cancer research trials carried out in LMICs, we discovered that of 35 breast cancer research trials carried out in the 3 LMICs of South Asia (India, Pakistan and Bangladesh), only 2 (5%) were located in Pakistan [4].

The ClinicalTrials.gov database is free to use and contains details of more than 460,000 privately and publicly funded clinical studies conducted in 221 countries [5]; an analysis of these studies can identify knowledge gaps and emerging trends in clinical trial research. This article describes an analysis and overview of the past, ongoing and planned clinical trials of various therapies with at least one participating site in Pakistan. We have also included a summary of some of the challenges facing this country and local initiatives that provide a “personalized” approach to the provision of healthcare. This information will be useful to a researcher planning to execute a clinical trial in Pakistan and provides suggestions for future studies, particularly for low-cost and repurposed drug therapies.

## 2. Materials and Methods

Specific search terms were used in the query section of ClinicalTrials.gov. The comma-separated values (CSV) file of all studies meeting the query criteria was downloaded and analyzed using Microsoft^®^ Excel^®^ for Microsoft 365 and JMP^®^ Pro 17.01.0 running on Windows 10.

## 3. Results

### 3.1. Analysis of Studies Reported in ClinicalTrials.gov

#### 3.1.1. Study Selection

A query of [Pakistan] in “location” on 14 August 2023 of 461,791 studies listed on ClinicalTrials.gov found 2564 studies, all of which were then taken forward for analysis. Figure 1 shows a flow diagram illustrating these steps. The data can be found in Appendix A.

#### 3.1.2. Geographic Distribution

The geographic distribution of the studies is listed in Table 1. Most studies (2187/2564 = 85%) had a single site in Pakistan only. A further 122 studies had a single site in Pakistan plus at least one site outside of Pakistan. In total, 8% (197/2564) of studies had at least one site outside of Pakistan.

#### 3.1.3. Status of Studies

The status of the studies is listed in Table 2. Most studies (1643/2564 = 64%) had “completed” (the study had ended normally, and the last participant’s last visit had occurred). In total, 533 studies were actively recruiting participants (“enrolling by invitation” or “recruiting”). One study was marked as “suspended” but may start again (NCT05305352). The 16 studies that had been “terminated” had stopped early and will not start again. The 10 studies marked as “withdrawn” stopped before enrolling the first participant. The 208 studies marked as “unknown” had a last known status of “recruiting”, “not yet recruiting” or “active and not recruiting” but have passed their completion dates, and the status had not been verified within the past two years.

Only 2.5% (63/2564) of the studies had results available.

#### 3.1.4. Start Date

The start dates of the studies are shown in Figure 2 and range from April 1991 (NCTNCT00358943) to December 2025 (NCT04218786). Note that studies can either report an estimated date on which the clinical study will be open for recruitment of participants (resulting in future start dates) or the actual date on which the first participant was enrolled. One study did not report a start date (NCT00538811). Most studies had a start date of 2020 or later (1673/2564 = 65%).

#### 3.1.5. Phases of Studies

The phases of the studies are listed in Table 3. For most studies, the phase was “not applicable” or “not specified”.

#### 3.1.6. Study Designs

Most studies were categorized as interventional (2210/2564 = 86%), and the majority were categorized as randomized (2021/2564 = 79%). A breakdown by intervention model and masking (blinding) is shown in Table 4.

#### 3.1.7. Study Participants

From the eligibility criteria, the age groups and sex of the participants are tabulated in Table 5. Most studies included adult men and women, and an appreciable proportion included children (728/2564 = 28%). Fewer than 1% of studies restricted eligibility to people 65 years and older (9/2564).

A histogram of the number of participants in each study is shown in Figure 3.

#### 3.1.8. Types of Intervention

Table 6 lists the types of interventions used in the studies. Drug interventions were reported in 40% of studies (1035/2564).

#### 3.1.9. Study Funder

Table 7 lists the funder types, with only 7% of studies reportedly funded by industry (178/2564).

## 4. Discussion

The analysis of clinical trials conducted with at least one site in Pakistan provides evidence of a flourishing and capable infrastructure. Of the 2564 studies identified, 85% had a single site in Pakistan only (Table 1). The pace of new studies seems to be accelerating (Figure 2), with 65% of studies having a start date of 2020 or later. About 79% of the studies were randomized (Table 4), which provides the best unbiased evidence for the safety and efficacy of interventions [6]. A wide variety of types of clinical trials have been and continue to be performed, covering a range of interventions, including behavioral, biological, medical devices, diagnostic tests, dietary supplements and of course, drugs (Table 6). Although the majority of studies have an enrolment of 100 participants or fewer, there are some very large studies, including a study conducted by the Aga Khan University in Karachi with an estimated enrolment of 5,000,000 women and children, a remarkably large number (NCT04184544).

As an illustration of the diversity of clinical trials, here is a brief description of six of the studies currently recruiting or about to open:Effect of Soleus Pushup Exercise in Type 2 Diabetes Patients (NCT06020846) sponsored by Riphah International University in Rawalpindi. An RCT designed to test whether a simple exercise intervention can lower blood glucose levels in people with Type 2 diabetes.Effects of Prosthesis Training on Pain, Prosthesis Satisfaction and Ambulatory Status of Lower Limb Amputees (NCT06013631) sponsored by the University of Lahore. An RCT assessing the effects of prosthesis training with and without phantom exercises on people who have had a lower-limb amputation.Unified Protocol for Transdiagnostic Treatment for Depression and Anxiety in Adults (NCT06002087) sponsored by the National University of Science and Technology in Islamabad. A pilot RCT of a culturally adapted Unified Protocol for transdiagnostic psychological treatment in adults with anxiety and/or depression to assess the feasibility and acceptability.Effects of Posterior–Anterior Vertebral Mobilization Followed by Prone Press-up Exercise in Nonspecific Low Back Pain (NCT05997069) sponsored by the Sindh Institute of Physical Medicine and Rehabilitation in Karachi. An RCT designed to test a non-pharmacological method to alleviate lower back pain.Effect of Preoperative Tramadol and Naproxen Sodium on Post Operative Pain (ID NCT05982392) sponsored by Dow University of Health Sciences in Karachi. A three-arm RCT comparing a single intervention of oral tramadol, oral naproxen or placebo with a primary outcome of pain after dental treatment. To reduce bias, the trial is quadruple-blinded (masked) to the participant, care provider, investigator and outcomes assessor.Impact of Bi-26 Supplementation on Weight Gain in Underweight Infants (NCT05952076) sponsored by the Bill and Melinda Gates Medical Research Institute. A placebo-controlled RCT conducted in three sites in Pakistan evaluated the effect of supplementation with a strain of a gut microbe *Bifidobacterium longum* on weight gain in underweight infants between 30 and 120 days of age.

Our study builds upon and extends previous work in this area. A study of 508 clinical trials conducted in Pakistan registered on clinicaltrials.gov from 1992 to 2019 found an increase in the number of studies registered over time [7]. Analysis restricted to studies registered on clinicaltrials.gov is a potential limitation. However, there is no public register in Pakistan of all clinical trials run within the country [8].

The 2018 healthcare expenditure for Pakistan’s estimated 250 million populace was $4.5 billion (Figure 4); for comparison, the 2018 UK healthcare expenditure is calculated to be $259 billion for a population of 67 million. Akin to many other LMICs, Pakistan faces significant challenges in its healthcare system. Key issues impacting healthcare delivery are (i) a rapidly growing population, (ii) a historically anemic economic growth rate and (iii) inadequate funding for healthcare. The population of Pakistan has nearly septupled since 1947 (from 35 to 227 million), but health expenditure as a percent of gross domestic product (GDP) has been stagnant at an average of 0.75% over this period (Figure 4, for comparison, health expenditure in the UK since the 1970s has been around 5% of GDP and currently stands at 8%). In 1960, the estimated expenditure on healthcare was $20 million for a population of 47 million and a GDP of $3.7 billion; by 1980, the GDP had increased by 7% over the preceding 20 years; however, the population nearly doubled during the same period and expenditure on healthcare was a measly $165 million (Figure 4). By 2018, the healthcare expenditure increased to nearly $5 billion with the total population nearly tripling compared to 1980 (Figure 4). These data indicate that although the expenditure on healthcare (as a percentage of GDP) has been comparatively low, albeit consistent, the near-exponential population growth has meant there is a real-term decline in healthcare funding and access. This is a significant difference that has both constrained major improvements in health metrics and accessibility to therapies that are taken for granted in developed countries. In addition, Pakistan’s healthcare system further grapples with numerous other serious issues, including a shortage of healthcare professionals and rural and urban disparities in healthcare services.

The healthcare system in Pakistan is a blend of public and private sectors, with a vast majority of the population relying on public healthcare services that are largely not fit for purpose. Healthcare delivery is jointly administered by the federal and provincial governments. The system can be divided into three tiers: primary, secondary and tertiary care. Primary healthcare facilities are the first point of contact for most patients and are designed to provide basic healthcare services. Secondary care facilities offer more specialized services and are generally located in larger cities. Tertiary care facilities, such as teaching hospitals, provide advanced medical care and training for healthcare professionals [10].

Challenges facing the health system include [11]:

Inadequate Funding: One of the most significant challenges is the insufficient allocation of funds to the healthcare sector. Pakistan’s expenditure on healthcare is well below the recommended levels set by international organizations, resulting in a lack of resources for infrastructure development, medical equipment and training of healthcare professionals.

Limited Access to Quality Healthcare: Access to quality healthcare services is a major concern, especially in rural and remote areas. Most healthcare facilities are concentrated in urban centers, leaving a significant portion of the population underserved. Healthcare disparities due to geographic location are a problem in many countries, including high-income countries, although there is evidence that Canada, Norway and the Netherlands have managed to reduce these disparities [12]. Clinical trials to test the application of methods used in these countries could be used to improve geographic-based health equity in Pakistan.

Shortage of Healthcare Professionals: Pakistan faces a shortage of trained healthcare professionals, including doctors, nurses and paramedics. This shortage is particularly severe in rural areas, where healthcare professionals are often reluctant to work due to inadequate facilities and security concerns.

High Disease Burden: Pakistan has a high burden of communicable diseases, such as malaria, tuberculosis and hepatitis, as well as non-communicable diseases, including cardiovascular diseases and diabetes. Controlling these diseases requires a robust public health infrastructure and preventive measures.

Maternal and Child Health: Maternal and child health indicators remain concerning, with high maternal mortality rates and under-five mortality rates. The lack of access to prenatal and postnatal care, skilled birth attendants and appropriate healthcare for children contribute to these issues.

Sanitation and Hygiene: Poor sanitation and hygiene practices in many parts of the country exacerbate health issues, leading to preventable diseases like diarrhea and cholera.

Despite these challenges, Pakistan has made some strides in recent years to improve its healthcare system.

National Health Vision 2025: The government launched the National Health Vision 2016–2025, which aims to revamp the healthcare system by improving infrastructure, increasing human resource capacity and enhancing access to quality healthcare services [13].

Universal Health Coverage Program: Pakistan introduced the Sehat Sahulat Program to provide health coverage to low-income families, offering financial protection against high medical costs [14].

Expanded Immunization Programs: The country has expanded its immunization programs to reach more children, increasing vaccination rates against preventable diseases [15].

Polio Eradication Efforts: Pakistan has made significant progress in eradicating polio, with a considerable decline in cases in recent years. However, challenges remain due to security issues in certain regions and vaccine hesitancy [16].

Digital Health Initiatives: The adoption of digital health initiatives, such as telemedicine and electronic health records, has shown promise in increasing healthcare access and efficiency, especially in remote areas [17].

To build a more robust and equitable healthcare system, Pakistan needs to implement strategic measures and reforms. Some key recommendations include:

Increased Funding: A key future objective should be an increase in healthcare expenditure, prioritizing healthcare and allocating adequate funds to strengthen the infrastructure, procure modern medical equipment and support research and development. For example, the economic burden of mental illness in Pakistan was estimated to be £2.97 billion ($3.74 billion) in 2020, highlighting the need to develop a cost-effective national model for mental healthcare [18].

Human Resource Development: Invest in training and retaining healthcare professionals, especially in underserved areas, by offering incentives, such as better salaries, improved working conditions and career development opportunities. For example, a study exploring the facilitators and barriers to the acceptability of volunteer laywomen from the community as delivery agents of a psychosocial intervention for perinatal depression in a rural area of Pakistan found that this could be a way of delivering an effective intervention in relatively low cost [19].

Addressing Inequalities: Address regional disparities by focusing on expanding healthcare facilities and services in rural and remote areas, where access is limited. A study of women seeking surgical care found evidence for gender disparity [20], particularly in marginalized and hard-to-reach groups [21].

Public Health Interventions: Prioritize preventive healthcare measures, such as promoting hygiene and sanitation, expanding immunization coverage and the education of ordinary citizens. For example, the Pakistan Life Savers Programme aims to empower people with the skills needed to save lives and has trained over 225,000 life savers [22]. A notable achievement is in the use of cardiopulmonary resuscitation tailored to low-resource settings, rather than simply adopting strategies used in high-income countries [23].

Maternal and Child Health: Strengthen maternal and child health programs by increasing access to skilled birth attendants, improving prenatal and postnatal care and promoting community-based interventions. For example, the implementation of pulse oximetry might improve child morbidity and mortality from pneumonia but requires improved access to pulse oximeters and oxygen supplies [24].

Health Awareness Campaigns: Launch public health awareness campaigns to educate the population about preventive measures, healthy lifestyles and disease management. For example, mass gatherings such as the Hajj pilgrimage may facilitate the spread of infectious diseases and antimicrobial resistance [25]; health awareness campaigns that have been demonstrated in clinical trials to be effective could potentially have a hugely beneficial impact on healthcare spending.

Public–Private Partnerships: Foster collaboration between the public and private sectors to leverage resources and expertise for improving healthcare delivery. For example, Punjab (the most populous region of Pakistan) has the most developed healthcare infrastructure in the country, due in no small part to the implementation of public–private partnerships [26].

Technology Integration: Continue to embrace digital health technologies to enhance healthcare access and efficiency, particularly in remote and underserved areas. For example, the use of remote health monitoring systems for elderly people could make use of data from wearable sensors collected through cloud servers and analyzed by AI to flag potential issues and alert their families to take preventive action [27,28]. Ideas such as these must be tested locally through clinical trials before implementation.

As outlined above, the healthcare system in Pakistan suffers from chronic and deeply entrenched economic, infrastructure and medical personnel crises. The structural problems are complex and deep-seated, and it is beyond the scope of this analysis and the expertise of the authors to propose any meaningful solutions. Even with determination, increased resources and concerted effort by its government, these problems will take many decades to resolve. We do, however, suggest that there may be one avenue, namely, drug repurposing, that could be pursued to address both the problems of population-tailored small molecule therapy and healthcare accessibility for a population in the short to medium term.

Drug repurposing (DR, also termed indication expansion, drug repositioning or re-profiling, i.e., discovering new uses for licensed drugs) has the potential to provide a cost-effective means of developing new therapies. This approach has numerous advantages in relation to LMICs in general and Pakistan in particular: (i) While de novo drug discovery is an expensive and time-consuming process, taking up to 15 years from concept to translation with a very high rate of attrition [29], DR can reduce the time, cost and risk; (ii) availability of abundant, real use, safety and drug interaction data; (iii) reduce time, risks and costs of clinical trials which can be initiated at Phase II stage. For example, treatment with repurposed therapies has been proposed for non-small-cell lung cancer [30] and Parkinson’s disease [31].

Notably, none of the 2564 studies we analyzed was investigating repurposed medications. This could be due to a lack of economic incentives, but “social finance” might be a way to solve this problem [32]. We propose that a nascent clinical trial effort in Pakistan for repurposed drugs for common ailments may help in the short to medium term to address severe deficiencies in healthcare disparities confronting the population in Pakistan.

## 5. Conclusions

Pakistan’s health system has undergone significant progress in recent years despite multiple challenges. The COVID-19 pandemic clearly illustrated the need for high-quality healthcare evidence, and the government’s commitment to healthcare reform and strategic initiatives are encouraging signs that Pakistan sees the need to invest in high-quality randomized clinical trials to ensure that the limited resources available are used most efficiently to find effective health interventions that are suitable for its unique circumstances. With a concerted effort to address funding gaps, improve access to quality healthcare including preventive measures and strengthen the healthcare workforce, Pakistan can achieve a more equitable and effective healthcare system, ultimately improving the well-being of its citizens.

## Figures and Tables

**Figure 1 jpm-13-01559-f001:**
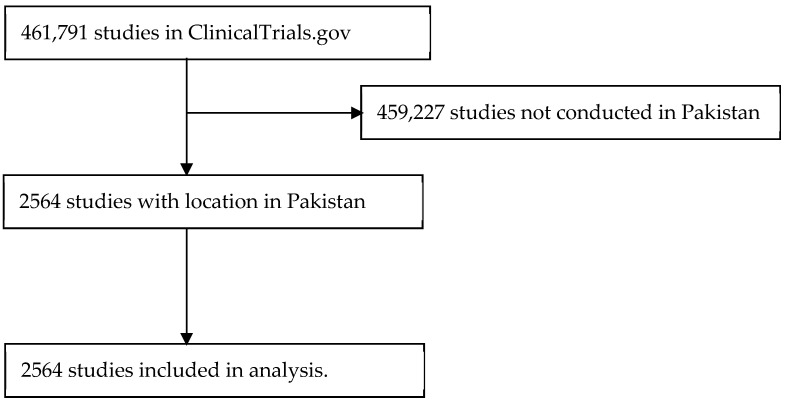
Selection of studies from ClinicalTrials.gov.

**Figure 2 jpm-13-01559-f002:**
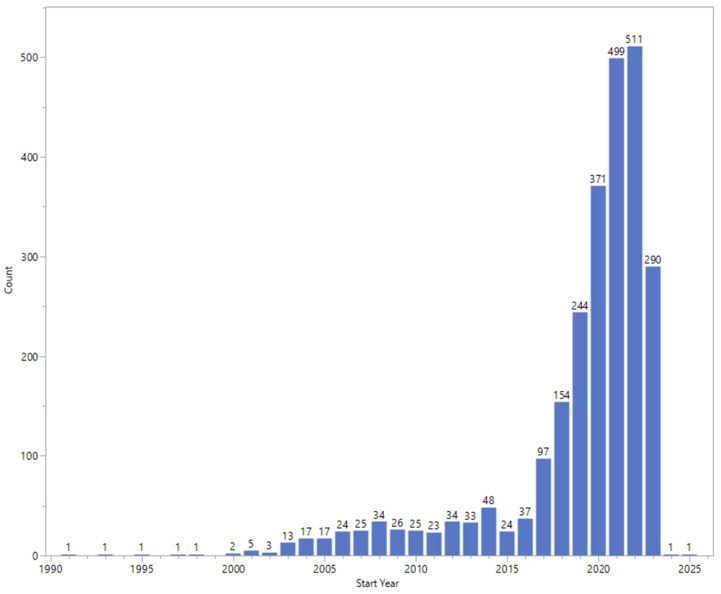
Start date (year) of the studies. Each column is labeled with the number of studies started (or expected to start) in that year.

**Figure 3 jpm-13-01559-f003:**
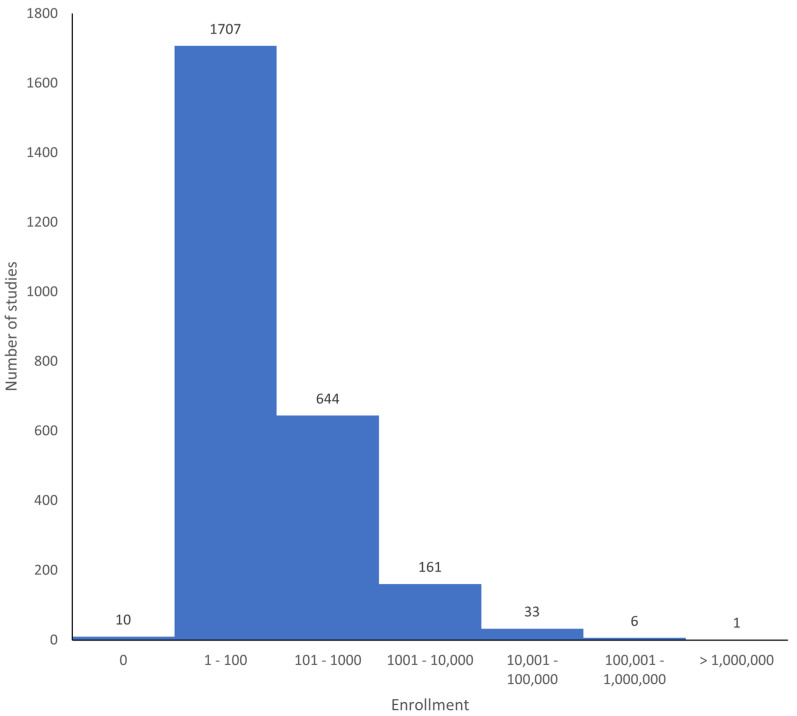
Numbers of participants (enrolment) in the studies. One study had an enrolment of 5,000,000 participants (NCT04184544). Two studies did not state the enrolment (NCT00327574 and NCT00198601).

**Figure 4 jpm-13-01559-f004:**
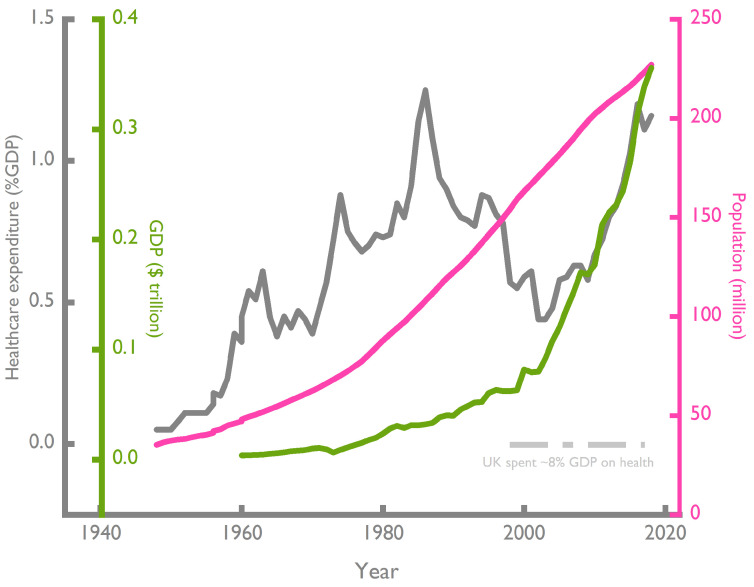
Data source (www.sbp.org, accessed on 31 August 2023) [9] for expenditure and World Bank for population growth [2]); World Bank sourced data from www.macrotrends.net accessed on 31 August 2023 for the GDP. For this figure, we consider 1 million = 1 × 10^6^, 1 billion = 1 × 10^9^ and 1 trillion = 1 × 10^12^.

**Table 1 jpm-13-01559-t001:** Distribution of locations of sites.

Number of Sites within Pakistan	Number of Studies with Sites only in Pakistan	Studies with Sites in Pakistan Plus at Least One International Site ^1^
1	2187	122
2	121	34
3	19	13
4	17	6
5	2	11
6	6	3
7	2	1
8	3	3
9	1	1
10	0	2
11	5	0
15	2	0
17	0	1
20	1	0
28	1	0
Total	2367	197

^1^ For example, 2187 studies had one site in Pakistan only; a further 122 studies had one site in Pakistan plus at least one site in another country.

**Table 2 jpm-13-01559-t002:** Study status.

Study Status	Number of Studies
Not yet recruiting	90
Enrolling by invitation	30
Recruiting	503
Active not recruiting	63
Completed	1643
Suspended	1
Terminated	16
Unknown	208
Withdrawn	10
Total	2564

**Table 3 jpm-13-01559-t003:** Phase of study.

Phase of Study	Number of Studies
Early Phase 1	29
Phase 1	51
Phase 1/Phase 2	20
Phase 2	84
Phase 2/Phase 3	32
Phase 3	145
Phase 4	129
NA ^1^	1720
Total	2210 ^2^

^1^ Phase Not Applicable, such as trials of devices or behavioral interventions. ^2^ Phase was not specified in 354 studies.

**Table 4 jpm-13-01559-t004:** Type of randomized study.

Type of Randomized Study	Number of Studies
Intervention Model: Crossover	39
Masking: None	25
Masking: Single	11
Masking: Double	1
Masking: Triple	2
Intervention Model: Factorial	51
Masking: None	14
Masking: Single	19
Masking: Double	9
Masking: Triple	3
Masking: Quadruple	6
Intervention Model: Parallel	1886
Masking: None	539
Masking: Single	850
Masking: Double	322
Masking: Triple	92
Masking: Quadruple	82
Masking: NS ^1^	1
Intervention Model: Sequential	7
Masking: None	1
Masking: Single	5
Masking: Quadruple	1
Intervention Model: Single Group	34
Masking: None	15
Masking: Single	12
Masking: Double	4
Masking: Quadruple	3
Intervention Model: NS ^1^	4
Masking: None	2
Masking: Double	1
Masking: NS ^1^	1
Total number of randomized studies	2021

^1^ Not Specified.

**Table 5 jpm-13-01559-t005:** Sex and ages of participants eligible for the study.

Age Group ^1^	All	Female Only	Male Only	Not Specified ^2^	Total
Adult	730	214	63	0	1007
Adult, older adult	736	69	14	1	820
Child	305	3	3	0	311
Child, adult	121	43	13	0	177
Child, adult, older adult	208	29	3	0	240
Older adult	9	0	0	0	9
Total	2109	358	96	1	2564

^1^ The age groups are child (birth–17), adult (18–64) and older adult (65+). ^2^ In one study (NCT04728776), the sex was not specified.

**Table 6 jpm-13-01559-t006:** Types of intervention.

Type of Intervention	Number of Studies
Behavioral	225
Biological	81
Combination product	33
Device	177
Diagnostic test	105
Dietary supplement	112
Drug	1035
Genetic	2
Other	2392
Procedure	296
Radiation	15
Total	4473 ^1^

^1^ Some studies included more than one intervention.

**Table 7 jpm-13-01559-t007:** Funder of study.

Funder Type	Number of Studies
Other ^1^	2237
Industry	178
Other Government	121
Networks	18
U.S. National Institutes of Health	4
U.S. Federal agencies	3
Individuals	2
Unknown	1
Total	2564

^1^ Includes universities and community-based organizations.

## Data Availability

All data used in this work can be found in Appendix A.

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
