# Peer review of "Clinical Trials and Therapeutic Approaches for Healthcare Challenges in Pakistan"

_jpm, 2023, doi:10.3390/jpm13111559_

Round 1
Reviewer 1 Report
Comments and Suggestions for Authors
Outstanding study on clinical trials in Pakistan and challenges facing the healthcare system in Pakistan. Minor comment concerning Acronyms :authors should describe the full name of the acronym at its first appearance in the text.
Author Response
Reviewer 1
Outstanding study on clinical trials in Pakistan and challenges facing the healthcare system in Pakistan.
We thank the reviewer for their praise.
Minor comment concerning Acronyms :authors should describe the full name of the acronym at its first appearance in the text.
We have amended the text at lines 26 and 31 to describe the acronyms GDP and LMICs at first use (thank you for pointing out our oversight!)
We have included a revised draft manuscript incorporating the changes from all reviewers (highlighted in yellow) so that the responses can be seen in context.
Reviewer 2 Report
Comments and Suggestions for Authors
The reader may consider that the analysis cover Pakistan but it is a picture restricted to Clinicaltrials.gov. Still not clear to me how the article connects to the personalized medicine!
Author Response
Reviewer 2
The reader may consider that the analysis cover Pakistan but it is a picture restricted to Clinicaltrials.gov.
The reviewer is correct to point out that the basis of our analysis is restricted to clinicaltrials.gov database. The reasons for this are both practical and validatory. There are no publicly available registers in Pakistan of clinical trials run within the country. Fortunately, clinicaltrials.gov has attained the status of a benchmark for clinical trials around the world. All meaningful trials, from around the world, are registered at this website. The clinicaltrials.gov is akin to the standard of GenBank (https://www.ncbi.nlm.nih.gov/genbank/) for gene sequencing. In addition, the purpose of our analysis was to perform an overview of the status of clinical trials research in Pakistan, and the information held in clinicaltrials.gov is particularly suited for this. We have added text at line 204 to clarify.
Still not clear to me how the article connects to the personalized medicine!
There are two ways to address this point. Firstly, RCTs are often criticised as producing answers that are only applicable to those participants who met the eligibility criteria (i.e. lack of generalisability). Including participants who are like the target population will enable therapies to be more readily “personalised” (treating patients based on their individual demographic, genomic or biological characteristics). There are several examples of different pharmacogenomics in various people around the world.
Secondly, another way of addressing this issue would be to suggest that personalized medicine also relates to the local healthcare provision infrastructure. Therapies tested in the in the developed world might not be “transferable” to the rest of the world due to different provision of expertise, imaging equipment, laboratory tests, etc.
To achieve the holy grail of personalized medicine (a reductionist idea, however, still reliant on population analysis), it is necessary to test therapies in localized populations in their native environments through well designed and executed clinical trials.
We have added text at line 53 for clarification.
We have included a revised draft manuscript incorporating the changes from all reviewers (highlighted in yellow) so that the responses can be seen in context.
Reviewer 3 Report
Comments and Suggestions for Authors
Dear authors
I reviewed your manuscript entitled "Clinical Trials and Therapeutic Approaches for Healthcare Challenges in Pakistan". I have many queries and suggestions.
- In introduction, provide abbreviation for LMIC at first time use,
- In Materials and Methods, you mentioned that specific search terms were used in the query section of ClinicalTrials.gov.
Write the exact search terms you used.
- Tables 3 and 5 are not necessary. It is enough to mention in the text.
- Line 91 - The start dates of the studies are shown in Figure 2 and range from 1991 to 2025.
Did you start your search from 1991 or was the clinical trial started in 1991 in Pakistan? Now we are in 2023. Why do you include till 2025 as start date?
- Tables 7 and 8 can be combined together (Age and gender)
- In discussion - "Although the majority of studies have an enrollment of 100 participants or fewer, there are some very large studies, including a study conducted by the Aga Khan University in Karachi with an estimated enrollment of 5,000,000 women and children [7]".
This is your study result. Why do you give reference here?
- Figure 4 is not clear. The details you mentioned in the text is not reflected in the figure. In the figure, it was denoted in trillion. You wrote in the text in billion. 1 trillion is 1000 billion.
- Starting from lines 207 in discussion, all these were general statements, not related to your study results. Even if you don't do this research, you can provide these statements or information. Compare your study results with other studies and provide discussion related to your study.
Author Response
Reviewer 3
Dear authors
I reviewed your manuscript entitled "Clinical Trials and Therapeutic Approaches for Healthcare Challenges in Pakistan". I have many queries and suggestions.
- In introduction, provide abbreviation for LMIC at first time use,
Done, please see line 31.
- In Materials and Methods, you mentioned that specific search terms were used in the query section of ClinicalTrials.gov. Write the exact search terms you used.
Ultimately, the search was very simple - A query of [Pakistan] in “location” (line 67). We initially envisaged that additional search terms would be required, but this was not necessary.
- Tables 3 and 5 are not necessary. It is enough to mention in the text.
Thank you. Done, please see text at lines 116 and 124.
- Line 91 - The start dates of the studies are shown in Figure 2 and range from 1991 to 2025.
Did you start your search from 1991 or was the clinical trial started in 1991 in Pakistan? Now we are in 2023. Why do you include till 2025 as start date?
There were no restrictions placed on dates; the first start date was 1991. Some studies have been registered prospectively (which is encouraged) with a start date in the future. We have clarified this at line 99
- Tables 7 and 8 can be combined together (Age and gender)
The Tables have been combined into (new) Table 5 at line 130.
- In discussion - "Although the majority of studies have an enrollment of 100 participants or fewer, there are some very large studies, including a study conducted by the Aga Khan University in Karachi with an estimated enrollment of 5,000,000 women and children [7]".
This is your study result. Why do you give reference here?
Thank you for this point. We have removed the reference and refer to the NCT number for consistency (line 167).
- Figure 4 is not clear. The details you mentioned in the text is not reflected in the figure. In the figure, it was denoted in trillion. You wrote in the text in billion. 1 trillion is 1000 billion.
In the figure, it was denoted in trillion. You wrote in the text in billion. 1 trillion is 1000 billion.
We do mean millions when referring to population numbers and billions (100 billion = 0.1 trillion) when referring to GDP. For example, Pakistan’s GDP was 259 billion in 2018 which is equal to 0.259 trillion (259/1000). We have added text to the figure legend to clarify (line 232).
- Starting from lines 207 in discussion, all these were general statements, not related to your study results. Even if you don't do this research, you can provide these statements or information. Compare your study results with other studies and provide discussion related to your study.
The intention of this section of the manuscript is to show some of challenges facing Pakistan (many of which are unique), and to try and illustrate how clinical trials can help determine the optimal approaches to be taken to ensure the population has access to affordable (and truly personalised) healthcare. The Journal of Personalised Medicine has global readership, and hopefully this will stimulate thinking about ways of finding solutions to similar problems in other countries. Regarding the point about referring to other studies, we have added text starting at line 202.
We have included a revised draft manuscript incorporating the changes from all reviewers (highlighted in yellow) so that the responses can be seen in context.
Round 2
Reviewer 2 Report
Comments and Suggestions for Authors
Thanks for your convincing arguments and corrections
Author Response
Thank you for all your comments and suggestions which have improved the manuscript.
Reviewer 3 Report
Comments and Suggestions for Authors
Still I am not convinced with figure 2. You mentioned in the text "The start dates of the studies are shown in Figure 2 and range from 1991 to 2025".
The figure depicts only till 2023, why not till 2025 as you mentioned in the text?
Author Response
Still I am not convinced with figure 2. You mentioned in the text "The start dates of the studies are shown in Figure 2 and range from 1991 to 2025".
The figure depicts only till 2023, why not till 2025 as you mentioned in the text?
We agree that this Figure is not satisfactory, so have changed the axis to show the most recent study date and have labelled the columns with the number of studies. We have also added more clarification to the text from line 99.
Thank you for all your comments and suggestions which have improved the manuscript.